Resource

# Human genomic DNA is widely interspersed with i-motif structures

Cristian David Peña Martinez [ID] [1,2], Mahdi Zeraati[1,2,7], Romain Rouet[1,2], Ohan Mazigi[1,2], Jake Y Henry [ID] [1,2], Brian Gloss[1,2], Jessica A Kretzmann[3], Cameron W Evans[3], Emanuela Ruggiero [ID] [4], Irene Zanin [ID] [4], Maja Marušič[5], Janez Plavec[5], Sara N Richter[4], Tracy M Bryan [ID] [6], Nicole M Smith[3], Marcel E Dinger[1,2,8], Sarah Kummerfeld [ID] [1,2] & Daniel Christ [ID] [1,2 ✉]

## Abstract

DNA i-motif structures are formed in the nuclei of human cells and are believed to provide critical genomic regulation. While the existence, abundance, and distribution of i-motif structures in human cells has been demonstrated and studied by immuno-fluorescent staining, and more recently NMR and CUT&Tag, the abundance and distribution of such structures in human genomic DNA have remained unclear. Here we utilise high-affinity i-motif immunoprecipitation followed by sequencing to map i-motifs in the purified genomic DNA of human MCF7, U2OS and HEK293T cells. Validated by biolayer interferometry and circular dichroism spectroscopy, our approach aimed to identify DNA sequences capable of i-motif formation on a genome-wide scale, revealing that such sequences are widely distributed throughout the human genome and are common in genes upregulated in G0/G1 cell cycle phases. Our findings provide experimental evidence for the widespread formation of i-motif structures in human genomic DNA and a foundational resource for future studies of their genomic, structural, and molecular roles.

**Keywords** i-motif; DNA Quadruplex Structures; Immunoprecipitation; Antibody; iMab
**Subject Categories** Chromatin, Transcription & Genomics; DNA Replication, Recombination & Repair

## Introduction

Unravelling the location of regulatory elements in human genomic DNA is critical for our understanding of genome architecture and function. I-motif structures and related guanine-rich G-quadruplex structures (G4s) have been identified as important regulatory elements in gene transcription, DNA replication, telomeric and centromeric regions, and have been implicated in a range of human conditions (Abou Assi et al, 2018; Bochman et al, 2012; Kendrick et al, 2014; Li et al, 2016; Takahashi et al, 2017; Wells, 2007; Zeraati et al, 2017). While there has been extensive research into G4 location and function (Balasubramanian et al, 2011; Chambers et al, 2015; Hansel-Hertsch et al, 2016; Lam et al, 2013; Marsico et al, 2019), iM formation is less well-studied (Tao et al, 2024). Unlike the canonical double-stranded B-form DNA, i-motif (iM) DNA is formed by hemi-protonated intercalated cytosine base pairs folded into a tetrameric structure (Fig. 1A) (Abou Assi et al, 2018). Although it had been evident for several decades that cytosine-rich sequences can form i-motif structures in vitro, the observations that the formation of the i-motif structure is dependent on acidic conditions (pH 5–6) had initially raised questions concerning their formation in cells (Bochman et al, 2012; Wells, 2007). However, more recently, it has become apparent that iM structures can exist at physiological pH under conditions of molecular crowding and negative DNA superhelicity (Li et al, 2016; Takahashi et al, 2017; Zeraati et al, 2017). More recently, iM structures that fold at neutral pH have also been identified (Chambers et al, 2015; Kendrick et al, 2014; Marsico et al, 2019), and the existence, abundance and distribution of iMs in the chromatin context of human cells have been characterised by NMR (Viskova et al, 2024), and immunofluorescence (King et al, 2020) and CUT&Tag (Zanin et al, 2023) using the iM-specific antibody iMab (Zeraati et al, 2018).

## Results

### Immunoprecipitation, biophysical validation, and distribution of iM structures

To establish a map of iMs across protein-depleted human genomic DNA, we first isolated DNA and generated fragments of 100–200 bp through DNA shearing. Following a heat-cool step to promote single-stranded tetrameric structures, immunoprecipitation of iM DNA was carried out using a high-affinity anti-iM

[1]Garvan Institute of Medical Research, Darlinghurst, Sydney, NSW 2010, Australia. [2]St Vincent's Clinical School, Faculty of Medicine, University of New South Wales, Kensington, Sydney, NSW 2010, Australia. [3]School of Molecular Sciences, University of Western Australia, Crawley, WA 6009, Australia. [4]Department of Molecular Medicine, University of Padua, 35121 Padua, Italy. [5]Slovenian NMR Centre, National Institute of Chemistry, SI-1000 Ljubljana, Slovenia. [6]Children's Medical Research Institute, Faculty of Medicine and Health, University of Sydney, Sydney, NSW 2145, Australia. [7]Present address: School of Biotechnology and Biomolecular Sciences, University of New South Wales, Kensington, Sydney, NSW 2010, Australia. [8]Present address: Faculty of Science, University of Sydney, Camperdown, NSW 2006, Australia. ✉E-mail: d.christ@garvan.org.au

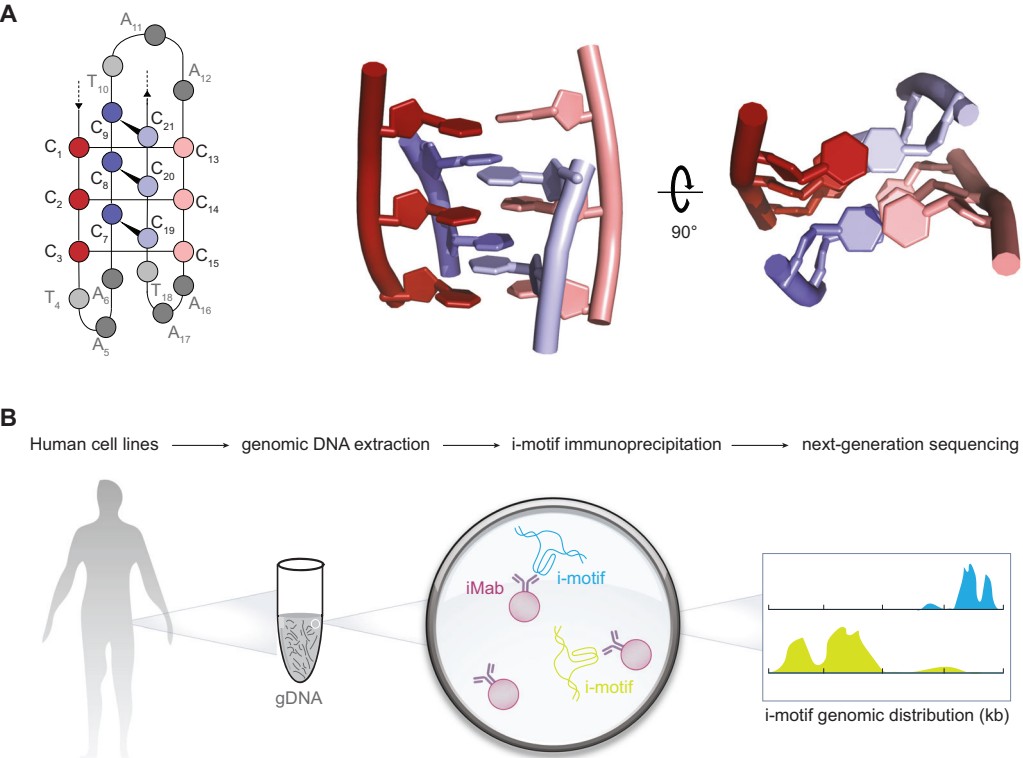

**Figure 1.  Identification of iM structures in human genomic DNA.**

(A) Schematic representation of intramolecular iM cytosine base pairing (C–C$^+$) and of canonical four-stranded iM structure (based on previously reported NMR structure (PDB: 1i9k)). (B) Immunoprecipitation and next-generation sequencing strategy used to identify iMs in human genomic DNA.

antibody (iMab) previously developed by our group (Zeraati et al, 2018) and physiological pH (pH 7.4) (Fig. 1B). Two independent biological replicates were carried out for each cell line to generate annotations of iMs across human genomic DNA. Immunoprecipitation steps were carried out at 4 °C (Fig. 2) (or alternatively at 16 °C as a control; Fig. EV1A) (Abou Assi et al, 2018; Zhou et al, 2010). We used DNA isolated from the MCF7 human breast cancer cell line (previously used for G4 studies (King et al, 2020; Lam et al, 2013)) as well as from U2OS osteosarcoma and HEK293T human embryonic kidney cell lines. Using this strategy, we identified 96,086 iM regions in MCF7 genomic DNA, 73,320 in U2OS and 86,826 in HEK293T, with high similarity in sequenced read count patterns observed between cell lines and replicates (Fig. EV1A–D). In total, 53,153 iM regions were observed among all three datasets (Fig. 2A,B).

To validate the sequences identified through immunoprecipitation and their ability to form iM structures, tetraplex structure formation was next analysed in vitro by biolayer interferometry and circular dichroism spectroscopy. For this purpose, we manually selected a set of 27 sequences located within the promoter regions of known oncogenes and tumour suppressor genes (Table EV1). The sequences were synthesised as DNA oligonucleotides and tested for binding to the iMab antibody (used for immunoprecipitation, as above) by biolayer interferometry: this confirmed that all of the analysed oligonucleotides bound to the iMab antibody with high-affinity and equilibrium binding constants ($K_D$) in the nM range (6–102 nM; Table EV1). Next, iM formation was further validated by circular dichroism

spectroscopy, with the majority of analysed sequences displaying spectra indicative of iM structures (Fig. EV2A, characterised by ~285 nm/~260 nm maximum/minimum (G Manzini and Xodo 1994; Wright et al, 2017)). For three of the iM sequences (*HOXC13, SIRPA,* and *TSHR)* folding was further investigated across a range of pH conditions (pH 5.0–8.0) with the analysed oligonucleotides displaying a high degree of pH-dependent folding (Figs. 2C and EV1E), a canonical feature of the iM structure (Day et al, 2013; Wang and Chatterton, 2021). Importantly, folding was observed at both 25 °C and at 37 °C, indicating the potential of the identified sequences for fold at physiological temperatures (Fig. EV2B).

We observed widespread distribution of iMs throughout the human genome, including in intergenic regions, introns, exons, and promoter regions (Figs. 1D and EV3A–C). Distances relative to the transcription start site (TSS) of genes were also highly variable with less than 30% observed within 10 kb of the TSS (Fig. 1E). Sequences were further analysed using the MEME software tool (Bailey et al, 2009), which revealed the enrichment of cytosine-rich motifs connected by thymidine tracts (Fig. 1F), in excellent agreement with the sequences of previously described iMs (Abou Assi et al, 2018; Fleming et al, 2018; Leroy, 2003; Školáková et al, 2019).

## Interplay of G4 and iMs

Prior experiments have suggested a close relationship between G4 and iM formation (Cui et al, 2016), with the dynamic formation of either structure having direct effect on the properties of the opposing DNA

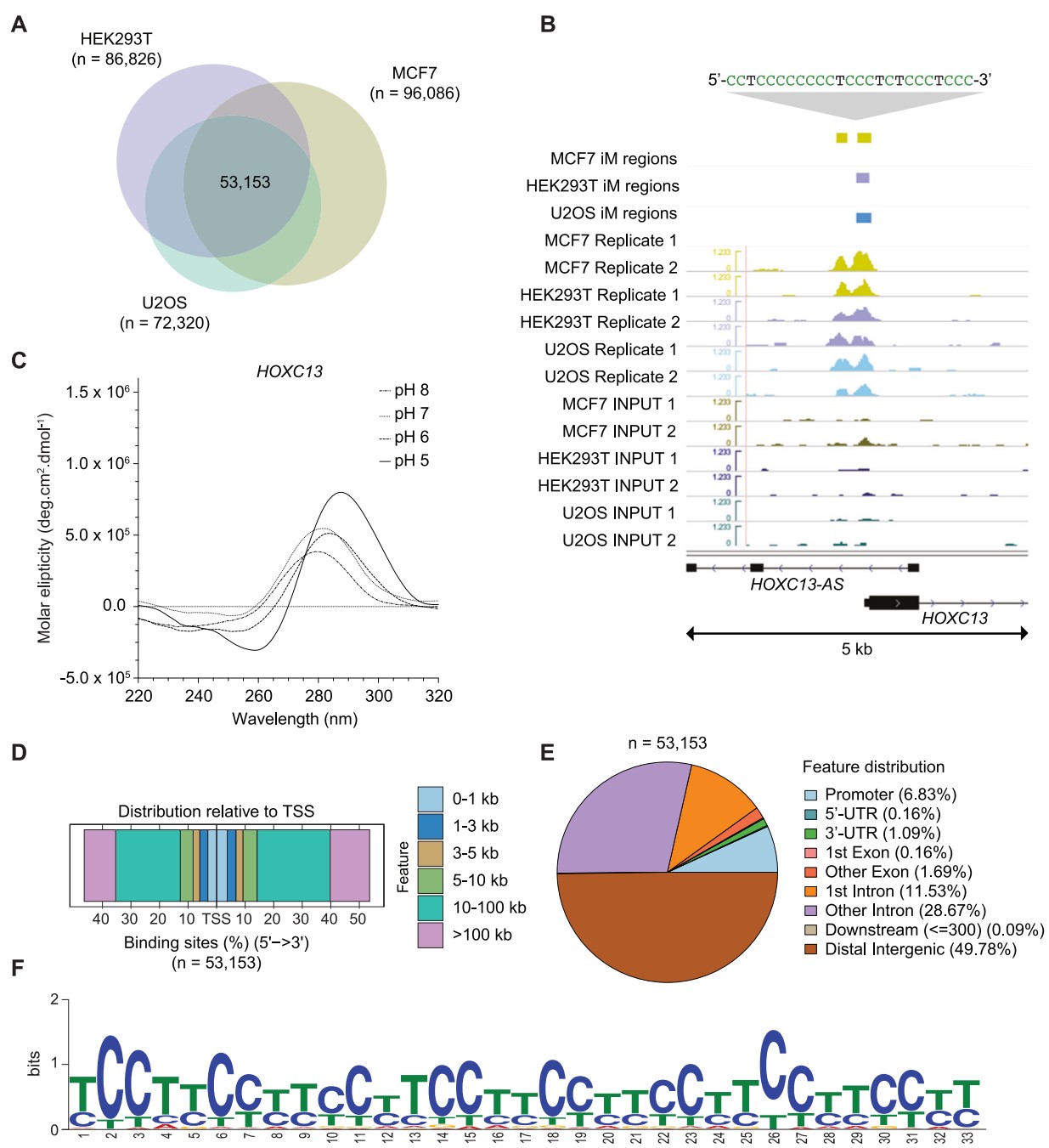

**Figure 2. iM structures are detectable and broadly distributed across human genomic DNA.**

(A) Total intersected iM regions observed after immunoprecipitation of protein-depleted purified DNA from three different human cell lines (53,153). Each cell line experiment was conducted twice using biological replicates. Coloured circles represent the regions intersected between cell line replicates (HEK293T; *n* = 86,826) (MCF7; *n* = 96,086) (U2OS; *n* = 72,320). (B) Genomic view highlighting an iM structure upstream of the *HOXC13* oncogene and downstream the transcription initiation site of *HOXC13-AS*. iM regions from each cell line replicate are shown (green tracks: MCF7, purple tracks: HEK293T, blue tracks: U2OS, lower tracks: control input profiles). (C) Validation of identified iM upstream of *HOXC13* by and circular dichroism spectroscopy under variable pH conditions (pH 5–8) and a temperature of 25 °C. (D) Distribution of iM structures across human genomic DNA. Percentage of genomic features. (E) Distribution relative to transcription starting sites. Represented regions (E, F) are the intersection across all three cell line experiments, *n* = 53,153. (F) Most frequently identified sequence motif observed in MCF7 DNA (MEME suite (Bailey et al, 2009)). NMR: Nuclear Magnetic Resonance, PDB: Protein Data Bank, MEME: Multiple Expectation maximisations for Motif Elicitation. All data shown from immunoprecipitated iMs at 4 °C and pH 7.4.

strand, leading to changes in genomic expression within neighbouring genes (Sun and Hurley, 2009). Using the iMab antibody utilised here, we have recently demonstrated that G4 and iM formation are interdependent and that the stabilisation of one structure can prevent the formation of the other on the opposing DNA strand (King et al, 2020). Locations of G4s in human genomic DNA have been previously reported in protein-depleted genomic DNA of MCF7 cells (using independent replicates and the small organic molecule pyridostatin (PDS) as a stabilising agent) (Chambers et al, 2015), allowing for comparison with the MCF7 iM dataset generated here. This analysis revealed considerable colocalisation between iM and G4 counts (Fig. 3A), indicating broad clustering of the two structures throughout DNA. Indeed, 71.6% (68,812/96,086) of iMs regions that were observed in both MCF7 biological replicates overlapped with previously reported G4s stabilised with PDS. This high level of colocalization was observed despite the previously reported G4 dataset being generated by polymerase stop assays (Chambers et al, 2015) rather than the direct immunoprecipitation technique utilised here (Fig. 3B).

Previous studies have further suggested the wide-ranging effect of iMs and G4s on transcription and gene regulation (Hansel-Hertsch et al, 2016; Miglietta et al, 2015; Varshney et al, 2020). To investigate this question, we determined their distance to the closest transcription starting site (TSS). For both DNA structures, we observed considerable overlap for protein-coding human genes (Fig. 3C), as indicated by high count frequency in relation to the TSS. However, we also noticed that iM regions were somewhat less frequent in the centre of the TSS compared to the previously reported G4 regions. The occupancy of iMs in relationship to the TSS is in agreement with biochemical studies of individual iMs (Brooks et al, 2010; Brown et al, 2017; Kendrick et al, 2014; Shu et al, 2018; Sun and Hurley, 2009) and computational predictions (Belmonte-Reche and Morales, 2020; Huppert and Balasubramanian, 2007; Kikin et al, 2006) that suggest a close correlation between tetrameric DNA formation and transcriptional regulation. In addition to transcription, G4 structures have been implicated in the control of replication (Besnard et al, 2012; Bochman et al, 2012). When analysing the previously reported G4 dataset, we observed a clear association of G4s with early (but not late) replication domains. However, such an association was not observed for the iM dataset, indicating different roles of the two DNA motifs in replication (Fig. EV3D) (Liu et al, 2016). In addition, our analyses revealed that iM regions are frequently observed in TAD boundaries that had previously been identified in MCF7 using Hi-C (Barutcu et al, 2015), with a majority (3174/32,273) of TAD regions containing iMs, and 24% (22,594/96,086) of iMs associated to a MCF7 TAD boundary region (Fig. EV3E).

## iMs and gene expression

The accessibility of chromatin to transcription factors (TFs) and the regulation of gene expression is mediated by DNA accessibility and histone modification, including methylation and acetylation (Bannister and Kouzarides, 2011; Lemon and Tjian, 2000; Vaquerizas et al, 2009). We examined genome coordinate overlaps in iM regions with data reported for multiple nuclear proteins in MCF7 cells from the ENCODE consortium, including histone modification markers, chromatin remodelling proteins, and DNA repair-associated proteins (2012; Manville et al, 2015; Nagarajan

**A**

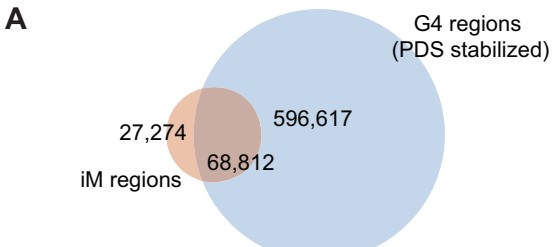

**B**

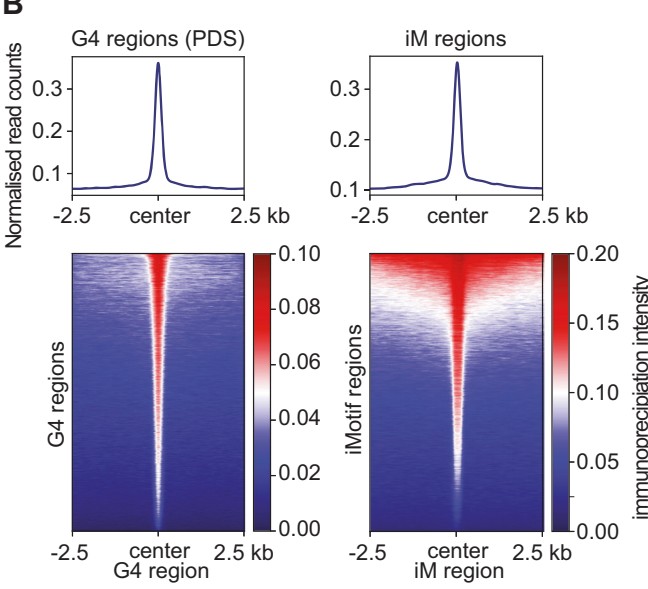

**C**

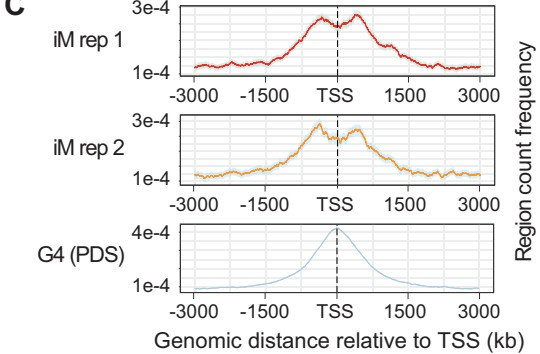

**Figure 3. Comparison of iM and G4 annotations.**

(**A**) Overlap of iM regions observed in protein-depleted DNA purified from MCF7 replicates and G4 regions previously reported (PDS stabilised) (Chambers et al, 2015). (**B**) Tag density histograms and heatmaps representing the occupancy of reads after iMab immunoprecipitation. Representative replicate from MCF7 purified DNA in proximity to published G4 regions stabilised by PDS (Chambers et al, 2015) (left panel) and occupancy of G4 reads in proximity to iM regions (right panel). Datasets are centred with 2.5 Kbp flanks. (**C**) Count frequency and distance (bp) of iMs and previously reported (PDS stabilised) G4s regions relative to TSS. MCF7 data shown from immunoprecipitated iMs at 4 °C and pH 7.4. PDS pyridostatin.

et al, 2014). Positive correlation of iMs with H3F3A, H3K27ac, H3K4me1, H2AFZ, H3K4me3, H3K4me2, and H3K9ac histone modifications were observed, indicating an association with gene transcriptional activity (Fig. EV4). A detectable correlation was also

observed for modifications associated with transcriptional elongation as represented by H4K20me1 (and contrasting with the H3K36me3 modification associated with heterochromatin, DNA repair and mitosis). In accordance with this, a negative correlation was also observed for H3K9me2 and H3K9me3, and a weak positive correlation with H3K27me3 histone modification marks, implying iM absence in genes with repressed transcription (Fig. EV4A). In contrast, comparison of the overlap of iM sites with those of chromatin remodelers suggests a more complex landscape of involvement in gene regulation, with positive correlations with the locations of MTA2, MTA3, SUZ12, and BMI1 proteins, which are associated with the control of repressed regions, whereas iMs did not correlate with other repressive chromatin remodelers such as SIN3A, CTBP1 and HCFC1 (Fig. EV4B). Overlap correlation and high global read counts at the locations of several TFs were further observed for iM regions, including E4F1, E2F8, POL2A, CLOCK, PAX8, GATAD2B, SP1 and CREB1 (Fig. EV4C). In contrast, iMs sites appear to not correlate with DNA repair-associated molecules (with the exception of ZBTB1; Fig. EV4A–D).

To further explore the association between iMs and gene expression, we surveyed bulk MCF7 RNA-seq. Our analyses revealed significant differences in the location of iM relative to a gene and its expression level. Genes with iMs associated to 5′ UTR, promoter-TSS (regions extending considerably from the TSS), 3′ UTR and exons displayed increased median mRNA levels (Fig. 4A), in agreement with the trend towards association of iMs with open chromatin and transcriptionally active DNA regions. As observed previously with a detectable decrease of iMs overlapping with the centre of the TSS region (Fig. 3C), genes with iMs at the TSS showed an overall probability of lower transcriptions levels than other groups (except non-coding regions). Overall, our result suggests that iMs positively correlate with genes with high mRNA expression rates (Fig. 4B), further supporting the notion that gene expression is regulated by iM formation. Ontology associations of iMs highlighted excitatory systems, including chemical channels of different tissues and extracellular signalling structures (Figs. 4C and EV5).

Finally, we analysed previously published nascent RNA (GRO-seq) sequencing datasets from MCF7 cells (Core et al, 2008; Liu et al, 2017), focusing on differential RNA-seq of upregulated and downregulated genes within G0/G1, S, and G2/M cell cycle phases. These analyses revealed an association of iMs and upregulated genes in the G0/G1 phase (G0/G1 vs G2/M) and (G0/G1 vs S); $P < 2.2 \times 10^{-16}$ while (G2/M vs S); $P = 0.735$] (Fig. 4D), consistent with previously reported data by our group and others that have demonstrated an increase of iM formation in G0/G1 phase by immunofluorescence (King et al, 2020; Zeraati et al, 2018).

## Discussion

While the existence of DNA iM structures in human cells in the context of chromatin has been demonstrated by immunofluorescent staining (Zeraati et al, 2018), and recently NMR (Viskova et al, 2024) and CUT&Tag (Zanin et al, 2023), specific insights into their location in protein-depleted human genomic DNA have so far been elusive. Here we use immunoprecipitation and high-throughput sequencing of protein-depleted human genomic DNA

from three different cells lines (MCF7, U2OS and HEK293T) to experimentally map sequences capable of iM formation in protein-depleted human genomic DNA. Our study further highlights the potential of the iMab antibody fragment for immunoprecipitation-based sequencing on a genome-wide scale, allowing the identification of a large number of DNA sequences capable of iM formation and their validation by biophysical characterisation in this study.

The use of the iMab antibody for the identification of iMs is further supported by three independent recent studies that used the antibody for immunoprecipitation, CUT&Tag and microarray analyses. More specifically, Ma et al (Ma et al, 2022) utilised immunoprecipitation to study iM-forming sequences in rice, which resulted in the identification of 25,306 iM sequences. Zanin et al, used the iMab antibody in CUT&Tag sequencing of human chromatin which identified 23,903 iM sequences in HEK293T cells (Zanin et al, 2023). Recently, Yazdani et al, analysed 10,976 DNA sequences derived from human promoters, centromeres and telomeres, which confirmed binding of iMab to previously validated iMs from genes including HRAS2, VEGFB and BRA (but not to a large set of control sequences) (Yazdani et al, 2023). While differences in experimental approaches exclude direct comparisons, taken together these results are in excellent agreement with the results reported here, which identified ~53,000 iMs among a set of three human cell lines (HEK293T, MCF7, U2OS).

In contrast to the published studies as above, recent work by Boissieras et al, using in vitro measurements reported that the iMab antibody binds to C-rich sequences independently of iM formation (Boissieras et al, 2024). This contrasts with the study by Zanin et al, (as well as the data outlined here), which demonstrates by CD that the identified sequences predominantly fold into iM structures (Zanin et al, 2023). Similar observations were also made by Ma et al, who reported that all the DNA sequences that were randomly selected from their set of 25,306 hits, and characterised by CD, formed iM structures (Ma et al, 2022) (the study was not referenced by Boissieras et al).

To further investigate the reported discrepancies, Ruggiero et al, have recently performed pulldown experiments using iMab and the set of synthetic oligonucleotides used by Boissieras et al, (preprint: Ruggiero et al, 2024). This analysis revealed that several of the oligonucleotides used by Boissieras et al, form *intermolecular* iM structures at higher DNA concentrations, which was further confirmed by NMR studies, providing a direct explanation for the observed discrepancies reported by the authors (preprint: Ruggiero et al, 2024). In contrast, highly iM-specific pulldowns were observed at lower DNA concentrations in the physiological range, highlighting the specificity of the iMab antibody, which recognises both intra- and intermolecular iMs (preprint: Ruggiero et al, 2024).

It is important to note that the observed differences are likely to partially reflect different experimental procedures, conditions, and reagents. Thus, immunoprecipitation studies including the studies by Ma et al (Ma et al, 2022) and study described here are carried out at 4 °C, while cellular studies are generally carried out 37 °C (Viskova et al, 2024; Zanin et al, 2023; Zeraati et al, 2018). Moreover, different conditions were used: in particular, strong blocking conditions (milk powder, superblock, spermidine, high ionic strength, salmon sperm DNA) were used here and by the above studies but not by Boissieras et al (Boissieras et al, 2024), which might well have contributed to the non-specificity reported by the authors (preprint: Ruggiero et al, 2024). Specific antibody formats used in the different studies may

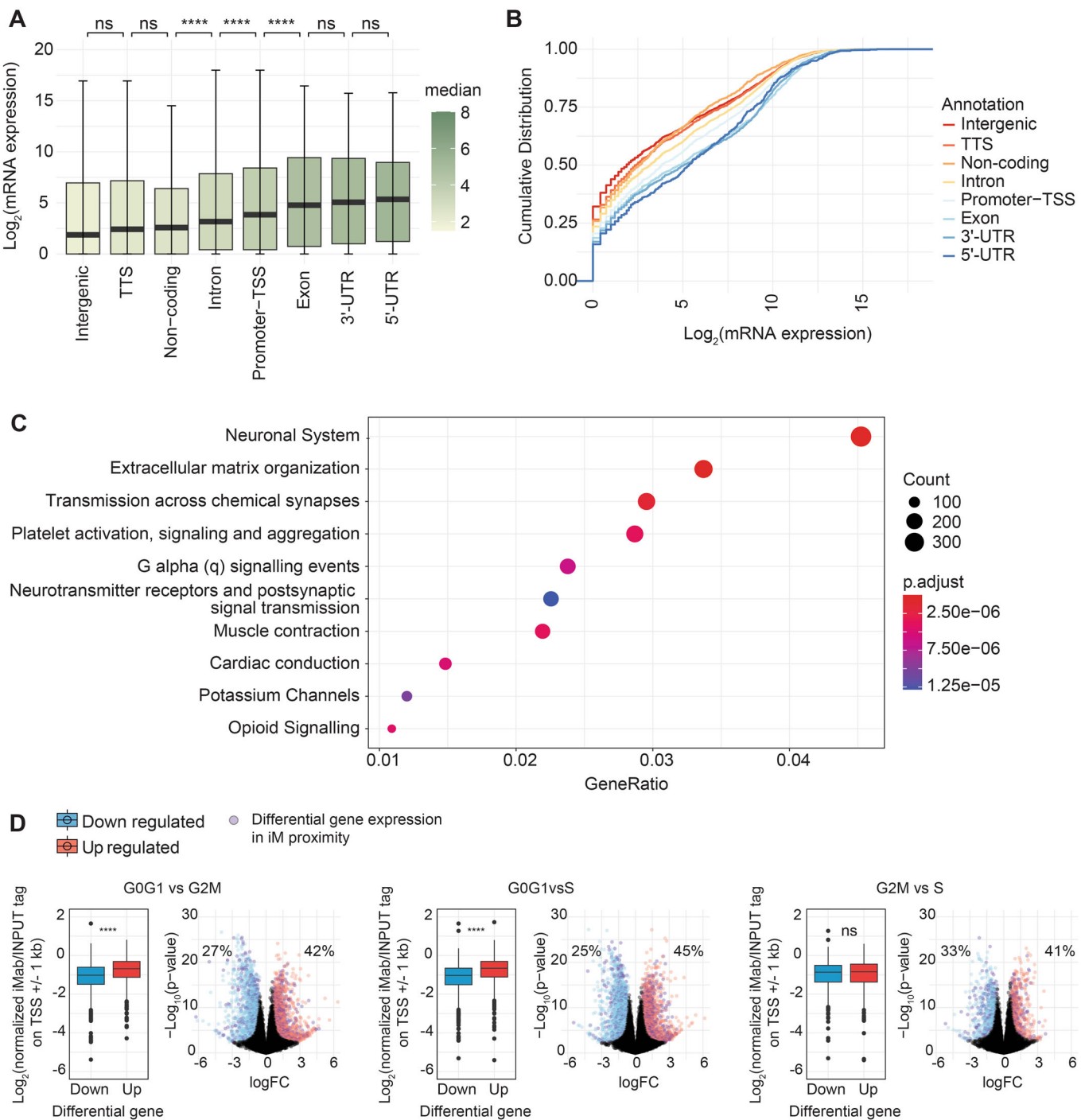

have also contributed to the observed differences, with some of the studies (including Ma et al) (Ma et al, 2022) utilising an IgG format, while other studies predominately used iMab in a single chain Fv (scFv) antibody format.

While differences in experimental approaches (including the use of rice DNA and acidic conditions (pH 5.5)) exclude direct comparisons, the studies by Ma et al, and Zanin et al, highlight the potential of iMab, developed in our laboratory, for immunoprecipitation and tagmentation approaches across a wide range of organisms. In contrast to the tagmentation studies above (Zanin et al, 2023), results outlined here

display a high level of consistency of iM regions among three different human cells lines (Fig. 2A), indicating experimental robustness and a high degree of iM convergence between cells of different morphological origins, including cancer cell lines. On the other hand, the iM mapping performed in the chromatin context, revealed different iM distribution among cell lines (Zanin et al, 2023), in line with the observation that the chromatin dynamic architecture influences cell identity (Klemm et al, 2019). Although the study outlined here utilised protein-depleted human genomic DNA, we observed overall similar results with the study of Zanin et al (Zanin et al, 2023): in both cases,

**Figure 4. iMs are distributed across human genes and are observed preferentially in the proximity of genes upregulated during early cell cycle phases.**

(A) Box plots of Log$_2$ (average mRNA expression) from three independent MCF7 cell bulk RNA seq experiments and the association with iMs regions at different human genomic annotations. The line representing the median and each boxplot is colour-coded for the median value and IQRs (interquartile range, box edges), with whiskers ranging at quantiles ± (1.5 × IQR). (TSS – intergenic; $P = 0.04$, intron—5′ UTR; $P = 0.006$, 3′ UTR—exon; $P = 0.003$), (Wilcoxon rank-sum test). (B) Cumulative distribution plot of iMab regions at different human genomic annotations. Plots represent the cumulative probability versus log$_2$ (average mRNA expression) of three independent bulk RNA experiments in MCF7 cells. (C) Reactome pathway enrichment analysis. Ten most significant pathways shown with $P$ value adjusted represented in a colour gradient and gene counts represented in the dot size. (D) Box plots and scatter plots of differentially expressed genes from nascent RNA (GRO-seq data: GSE94479 (Liu et al, 2017)) between G0/G1 vs G2/M, G0/G1 vs S, or G2/M vs S cell cycle phases in MCF7 cells and their relation against iMab reads near each gene transcription initiation site. Box plots for each differential analysis show upregulated and downregulated genes. Box plots indicate Log$_2$(normalised iMab/INPUT tag on TSS ±1 kbp) median values and IQRs with whiskers and outstanding data represented as points. Statistical significance was determined by the Wilcoxon rank-sum test between groups. (G0/G1 vs G2/M) and (G0/G1 vs S); $P < 2.2 \times 10^{-16}$ while (G2/M vs S); $P = 0.735$ Differential expression data (Volcano plots): log$_2$ FC (fold change) on the $x$ axis; $y$ axis log$_{10}$ ($P$ value). Volcano plots show genes in purple dots which have iMs regions in proximity to the gene body, including TSS, 5′ UTR, promoter, 3′ UTR and exon-related annotations. Percentages show the ratio of genes with proximal iMs over total differentially expressed genes in each group (upregulated or downregulated). Differentially expressed genes (statistically significant) with log$_2$ FC > 0.5 (G0/G1 vs G2/M; $n = 1587$ upregulated and $n = 1430$ downregulated genes) (G0/G1 vs S < 2; $n = 1951$ upregulated and $n = 1623$ downregulated) (G2/M vs S < 2; $n = 734$ upregulated and $n = 571$ downregulated genes).

iM structures were observed located in close proximity to G4 forming regions and in genes with high transcription rates, indicating that iM and G4 forming sequences in the human genome are mostly observed within regions that regulate gene expression.

We conclude that iM structures are located in close proximity to G4 forming regions, genes with high transcription rates, and those expressed in G0/G1 phase, highlighting their non-random distribution and involvement in genomic architecture. Our study provides foundational knowledge and resources relating to the location and distribution of iMs in human genomic DNA, representing potential targets for future diagnostic and therapeutic strategies.

# Methods

### Reagents and tools table

| Reagent/resource | Reference or source | Identifier or catalogue number |
|---|---|---|
| **Experimental models** | | |
| MCF7 cell line | Garvan Institute | Authenticated independently by analysis of microsatellite profiles in PCR amplified DNA extracted from the cells; short tandem repeat (STR) validation was performed using references from ATCC/DSMZ |
| HEK293T cell line | Garvan Institute | Authenticated independently by analysis of microsatellite profiles in PCR amplified DNA extracted from the cells; short tandem repeat (STR) validation was performed using references from ATCC/DSMZ |
| U2OS cell line | CellBank Australia | 92022711 |
| **Recombinant DNA** | | |
| pET12a | Garvan Institute | |

| Reagent/resource | Reference or source | Identifier or catalogue number |
|---|---|---|
| **Antibodies** | | |
| iMab scFv-Hisx6-FLAGx3 | Garvan Institute | |
| **Oligonucleotides and other sequence-based reagents** | | |
| DNA oligonucleotides | Integrated DNA Technologies | Table EV1 |
| **Chemicals, enzymes and other reagents** | | |
| DPBS | ThermoFisher | 14190144 |
| Dulbecco's Modified Eagle Medium (DMEM, high glucose, GlutaMAX™ Supplement) | ThermoFisher | 10566016 |
| SuperBlock™ Blocking Buffer | ThermoFisher | 37515 |
| **Software** | | |
| MEME v 5.3.3 | Bailey et al, 2009 | |
| R version 4.0.4 | R Core Team, 2021 | |
| bedtools | Quinlan and Hall, 2010 | |
| deepTools version 3.5.0 | Ramírez et al, 2016 | |
| Cutadapt version 3.5 | Kechin et al, 2017 | |
| FastQC software (https://www.bioinformatics.babraham.ac.uk/projects/fastqc/) | Andrews, 2010 | |
| bowtie2 Version 2.4.5 | Langmead et al, 2019 | |
| Samtools Version 1.10 | Danecek et al, 2021 | |
| macs2 Version 2.2.7.1 | Zhang et al, 2008 | |
| bedops Version 2.4.39 | Neph et al, 2012 | |
| picard Version 2.25.6 | http://broadinstitute.github.io/picard/ | |
| HOMER Version 4.11.1 | Heinz et al, 2010 | |
| Prism version 9 | GraphPad Software, Boston, Massachusetts USA, www.graphpad.com | |
| BLItz Pro software version 1.3. | ForteBio | |
| **Other** | | |
| Genomic DNA from cells was harvested using an AllPrep DNA/RNA/miRNA Universal Kit | QIAGEN | 80224 |
| Amicon Ultra-0.5 Centrifugal Filter Unit | Millipore | UFC501096 |
| ssDNA/RNA Clean & Concentrator kit | Zymo | D7011 |

| Reagent/resource | Reference or source | Identifier or catalogue number |
|---|---|---|
| AFA Fibre Pre-Slit Snap-Cap microtubes | Covaris | 520045 |
| M220 Focused-ultrasonicator, Covaris instrument | Covaris | |
| 96-well MaxiSorp plate | Thermo Scientific | 442404 |
| No-shearing IDT xGEN cfDNA & FFPE DNA | Integrated DNA Technologies | |
| 4200 TapeStation instrument | Agilent Technologies | |
| D1000 ScreenTapes | Agilent Technologies | 5067-5582 |
| NovaSeq 6000 SP reagent kit | Illumina | 20028312 |
| Illumina NovaSeq 6000 | Illumina | |
| Aviv 215 S circular dichroism spectrometer | Aviv | |
| BLItz system | ForteBio | |

## Antibody production

Expression and purification of the iM-specific antibody (iMab) was performed as previously described (Rouet et al, 2012; Zeraati et al, 2018). In brief, the iMab gene was cloned into pET12a vector encoding C-terminal c-Myc and Avi tags. Competent *E. coli* BL21-Gold (DE3) cells were transformed with pET12a-iMab and pBirAcm (Avidity). In vivo biotinylated iMab-MycAviTag antibody fragments were purified by affinity chromatography and biotinylation confirmed by biolayer interferometry using a BLItz system (Pall ForteBio LLC) and streptavidin sensors.

## Cell lines

MCF7 and HEK293T cell lines were authenticated independently by analysis of microsatellite profiles in PCR amplified DNA extracted from the cells; short tandem repeat (STR) validation was performed using references from ATCC/DSMZ. U2OS cell line was purchased from CellBank Australia. Cell lines were independently tested for mycoplasma contamination.

## DNA immunoprecipitation and sequencing

Cell lines were cultured in Dulbecco's Modified Eagle Medium (DMEM) + 10% v/v foetal bovine serum (FBS) and supplemented with penicillin-streptomycin antibiotics. Frozen cell line cultures were initiated from a single vial and split accordingly for biological replicates after the second passage. Genomic DNA from cells was harvested using an AllPrep DNA/RNA/miRNA Universal Kit (QIAGEN, Cat No: 80224) following the manufacturer's protocol. Purified DNA was concentrated using an Amicon Ultra-0.5 Centrifugal Filter Unit (Millipore, UFC501096) and concentration adjusted. DNA was fragmented using AFA Fibre Pre-Slit Snap-Cap microtubes (Covaris, 520045) and a Covaris instrument (M220 Focused-ultrasonicator) at Peak power: 50, Duty factor: 20, Cycles/Burst: 200, Time: 700 s, Setpoint: 20 °C producing 100–200 bp fragments. DNA fragments were stored at −80 °C until a pull-down experiment. A 96-well MaxiSorp plate (Thermo Scientific, 442404) was coated by adding 60 µL per well of 50 µg mL$^{-1}$ streptavidin diluted in PBS and incubated overnight at 4 °C. Fragmented DNA was thawed

and diluted in DPBS (ThermoFisher, 14190144), pH adjusted to 7.4. For each biological replicate, 140 µg DNA were diluted in 525 µL DPBS and heated at 90 °C for 10 min followed by cooling down at the rate of 1 °C per min to 21 °C and kept on ice until the pull-down experiment. A streptavidin-coated plate was washed once with PBS and blocked with 200 µL per well of SuperBlock (ThermoFisher, 37515) for 2 h at room temperature Biotinylated iMab-MycAviTag was diluted to 49 µg in 525 µL of DPBS pH 7.4 (7 wells, 75 µL each) for each biological replicate. Blocked wells were washed once with PBS and Biotinylated iMab-MycAviTag solution was added. After incubating for 1 h at room temperature while shaking at 200 rpm, wells were washed twice with DPBS pH 7.4. Then, 75 µL of DNA solution was added to each well and incubated 16–20 h at 4 °C, and additionally, in the case of the MCF7 cell line, two independent control experiments were also carried out at an incubation temperature of 16 °C. The supernatant was aspirated, each well-washed and tap-dried seven times with 200 µL of DPBS pH 7.4 supplemented with 0.1% v/v Tween 20 and twice with DPBS only at controlled room temperature of 24 °C. Elution was performed by adding 100 µL of fresh 100 mM Tris/acetate pH 10 supplemented with 1% w/v SDS to each well and incubating for 1 h at 40 °C while the plate was shaking at 200 rpm. Eluted DNA was collected, combined, and concentrated using ssDNA/RNA Clean & Concentrator kit (Zymo, Cat No.: D7011) following the manufacturer protocol. A no-shearing IDT xGEN cfDNA & FFPE DNA library preparation was performed to obtain libraries. A 4200 TapeStation instrument (Agilent Technologies) and D1000 ScreenTapes (Agilent Technologies, Part No: 5067-5582) with D1000 Reagents (Agilent Technologies, Part No: 5067-5583) were used for library quality controls. Sequencing was performed on NovaSeq 6000 (Illumina) using NovaSeq 6000 SP reagent kit in a paired-end 300 cycles mode following the manufacturer protocol.

## iM analysis

Paired reads were trimmed using trim_galore (Cutadapt version: 3.5) (Kechin et al, 2017). Quality control was performed on the trimmed reads using FastQC software (https://www.bioinformatics.babraham.ac.uk/projects/fastqc/). Reads were aligned to the hg19 reference genome using bowtie2 (Version: 2.4.5). Samtools (Version: 1.10) was used to obtain compressed binary files (.bam), sort and fix mates when necessary. Duplicates were marked using picard MarkDuplicates (Version: 2.25.6). Files were indexed using Samtools (Version: 1.10) and Bigwig files generated using bamCoverage (deeptools Version: 3.5.1) with the following parameters: -bs 10 -p 10 –normalizeUsing CPM. A list of high signal regions was blacklisted using the ENCODE's blacklist hg19-blacklist.v2 file (Amemiya et al, 2019) and a bedtools intersect command Regions were called using macs2 (Version: 2.2.7.1) using default parameters and INPUT bam files as the controls for each experiment. Intersecting regions were selected using bedops --intersect (Version: 2.4.39). Peaks were annotated using HOMER (v 4.11.1) (Heinz et al, 2010) and selected regions transformed back to fasta using bedtools -getfasta from reference genome hg19, sequences under 30 bp discarded using seqkit and de novo motif discovery was performed using MEME (v 5.3.3) (Bailey et al, 2009).

## DNA synthesis and biolayer interferometry

iM DNA sequences were synthesised and HPLC purified (Integrated DNA Technologies) and resuspended at 100 µM in nuclease-

free ddH$_2$O (ThermoFisher, cat No: 10977015). Binding kinetics of iMab and iM DNA oligonucleotides were determined by biolayer interferometry using a BLItz system (ForteBio) at room temperature. Biotinylated iMab scFv (or DP47/DPK9 as control) in 20 mM Tris-acetate pH 6.0 supplemented with 100 mM KCl was used to load the streptavidin biosensor for 2 min. Refolded DNA oligonucleotides 100 nm (in 20 mM Tris-acetate pH 6.0 with 100 mM KCl) were then used for association for 4 min, followed by 4 min dissociation in 20 mM Tris-acetate pH 6.0 with 100 mM KCl. Curve fitting was performed using BLItz Pro software version 1.3.

### Circular dichroism spectroscopy (CD)

DNA oligonucleotides were diluted to 10 µM in 20 mM Tris-acetate pH 6.0 with 100 mM KCl and annealed by heating at 90 °C for 10 min before cooling to room temperature at the rate of 1 °C min$^{-1}$. CD spectra were recorded using an Aviv 215 S circular dichroism spectrometer equipped with a Peltier temperature controller at 25 °C. Four scans were gathered over the wavelength ranging from 220 to 320 nm in a 0.1 cm path length cell and bandwidth 1 nm. A buffer-only blank spectrum was subtracted from the average scans for each sample. CD spectra were normalised to the total species concentrations. CD spectra were plotted using 4th order smoothing (10 neighbours) in GraphPad Prism. For pH gradient folding, DNA oligonucleotides were diluted to 10 µM in 20 mM sodium cacodylate at pH 5.0 to pH 8.0 with 100 mM KCl, and annealed by heating at 90 °C for 10 min before cooling to room temperature at a rate of 1 °C min$^{-1}$, as previously described (Wright et al, 2017).

### Analyses of G4-seq and ChIP-seq data

G4-seq data and multiple datasets from ChIP-sequencing, ATAC and DNase seq from MCF7 cells were obtained from previously published data (see Table EV2) and analyses performed using bedtools (Quinlan and Hall, 2010) and deepTools (Ramírez et al, 2016) version 3.5.0; analyses used regions within ± 1 kbp from the TSS of all refseq (O'Leary et al, 2016) genes, defined as TSS proximal iM regions. Read heatmaps and correlation plots were generated using deepTools (Ramírez et al, 2016) and R version 4.0.4. Ontology analyses were carried out in R using ChIPseeker (Yu et al, 2015), TxDb.Hsapiens.UCSC.hg19.knownGene, EnsDb.Hsapiens.v75, clusterProfiler, AnnotationDbi, and org.Hs.eg.db.

### Analyses of RNA-seq and GRO-seq data

RNA-seq and GRO-seq read counts were obtained from previously published data (see Table EV2) on non-hormone-stimulated MCF7 cells (Janky et al, 2014; Liu et al, 2017). Reads were annotated, depleted for low expressing genes (CPM < 0.5), normalised for composition bias and merged with iMab tag on TSS ± 2.5 kbp or tested for differential expression analysis. All reanalysis was conducted in R using the packages: EdgeR, limma, Glimma and org.Hs.eg.db.

### Statistical analyses

Statistical tests were carried out using deepTools (Ramírez et al, 2016), R version 4.0.4 and Prism 9. The ranges of $x$ and $y$ axes for scatter plots are justified in the figure captions. Statistical tests,

number of experiments and $P$ values are cited in figures and/or captions.

## Data availability

Sequencing data generated in this manuscript can be accessed through the Gene Expression Omnibus under the accession code GSE248445 for all DIP-seq immunoprecipitation data and accession code GSE248441 for MCF7 bulk RNA-seq results. All previously reported GEO datasets used for the analyses are listed in Table EV2.

The source data of this paper are collected in the following database record: biostudies:S-SCDT-10_1038-S44318-024-00210-5.

## Peer review information

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

## Acknowledgements

DC acknowledges funding from the National Health and Medical Research Council (1157744). The authors acknowledge the use of the services and facilities of the Australian Genome Research Facility (AGRF) and the facilities of the Children's Medical Research Institute (CMRI).

## Author contributions

**Cristian David Peña Martinez**: Conceptualisation; Data curation; Software; Formal analysis; Investigation; Visualisation; Methodology; Writing—original draft; Writing—review and editing. **Mahdi Zeraati**: Conceptualisation; Methodology; Writing—review and editing. **Romain Rouet**: Conceptualisation; Data curation; Formal analysis; Validation; Investigation; Methodology. **Ohan Mazigi**: Validation; Writing—review and editing. **Jake Y Henry**: Project administration; Writing—review and editing. **Brian Gloss**: Data curation; Software; Writing—review and editing. **Jessica A Kretzmann**: Writing—review and editing. **Cameron W Evans**: Writing—review and editing. **Emanuela Ruggiero**: Validation; Writing—review and editing. **Irene Zanin**: Validation; Writing—review and editing. **Maja Marušič**: Validation; Writing—review and editing. **Janez Plavec**: Validation; Writing—review and editing. **Sara N Richter**: Validation; Writing—review and editing. **Tracy M Bryan**: Resources; Formal analysis; Writing—review and editing. **Nicole M Smith**: Writing—review and editing. **Marcel E Dinger**: Conceptualisation; Writing—review and editing. **Sarah Kummerfeld**: Software; Formal analysis; Methodology; Writing—review and editing. **Daniel Christ**: Conceptualisation; Supervision; Funding acquisition; Methodology; Writing—original draft; Writing—review and editing.

Source data underlying figure panels in this paper may have individual authorship assigned. Where available, figure panel/source data authorship is listed in the following database record: biostudies:S-SCDT-10_1038-S44318-024-00210-5.

## Disclosure and competing interests statement

The authors declare no competing interests.

# Expanded View Figures

**Figure EV1.  iM immunoprecipitation (controls, repeats).**

(**A**) Regions observed across replicate experiments, intersected iM regions observed after immunoprecipitation of protein-depleted purified DNA from the MCF7 cell-line protein-depleted DNA. Immunoprecipitation experimental repeats using an incubation temperature of 16 °C or 4 °C. (**B**) Regions observed across experiment repeats in DNA from two different cell-lines. Replicates for HEK293T and U2OS protein-depleted DNA from pulldowns conducted at 4 °C. (**C**) Pairwise intersection-fraction of overlap pie charts of all-vs-all experiments conducted. (**D**) Genomic view highlighting an iM structure upstream of the oncogenes *ATM*, *SIRPA*, and *TSHR*. iM regions from each cell line replicates are shown (green tracks: MCF7, brown tracks: MCF7 DNA incubated at 16 °C, purple tracks: HEK293T, blue tracks: U2OS, lower tracks: immunoprecipitation negative control input profiles). (**E**) hTelo positive control CD curve under variable pH conditions (left panel) and validation of an identified iM candidate upstream of *SIRPA* and *TSHR* by DNA synthesis following CD spectroscopy under variable pH (pH 5–8) and a temperature of 25 °C. CD Circular Dichroism.

▶

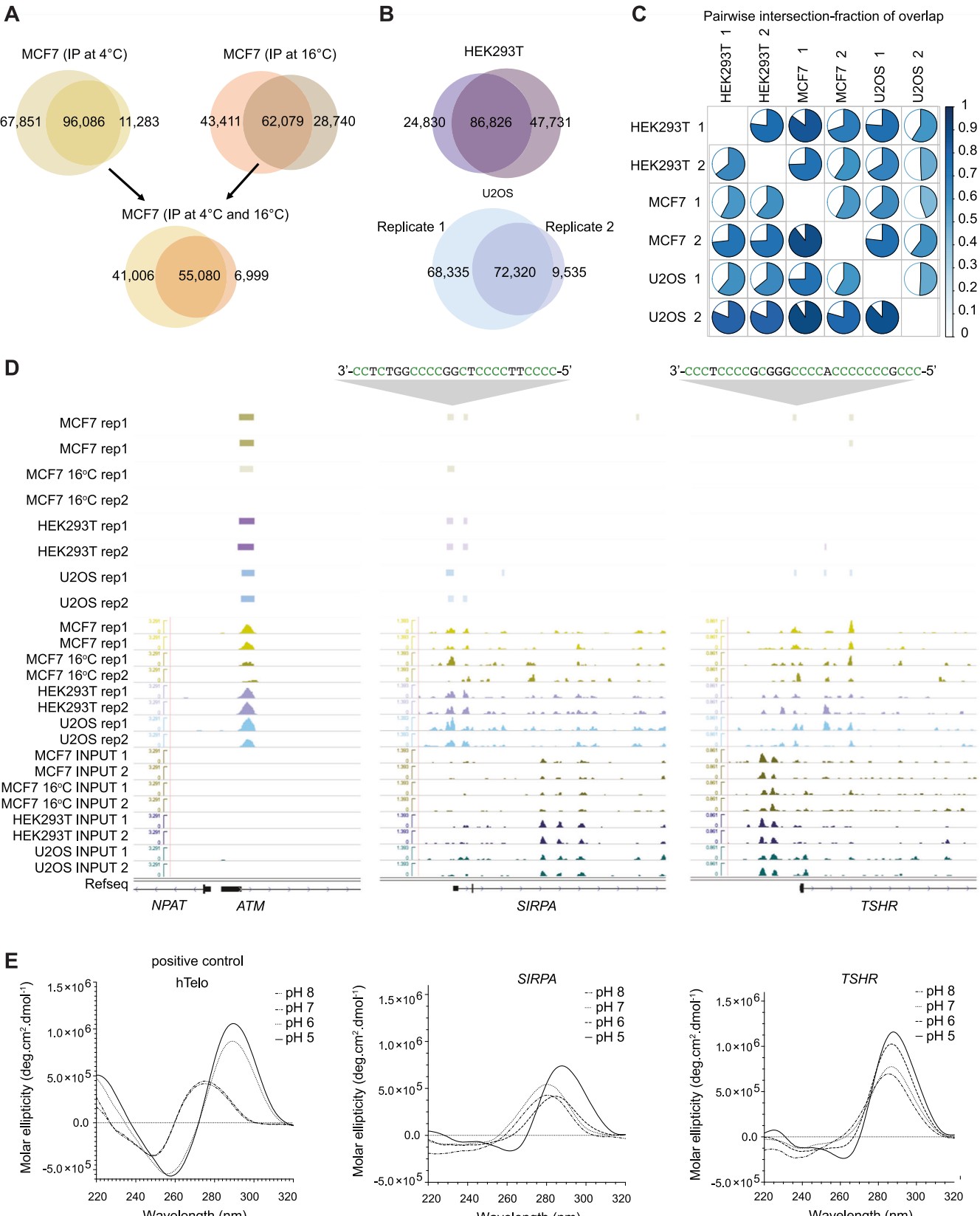

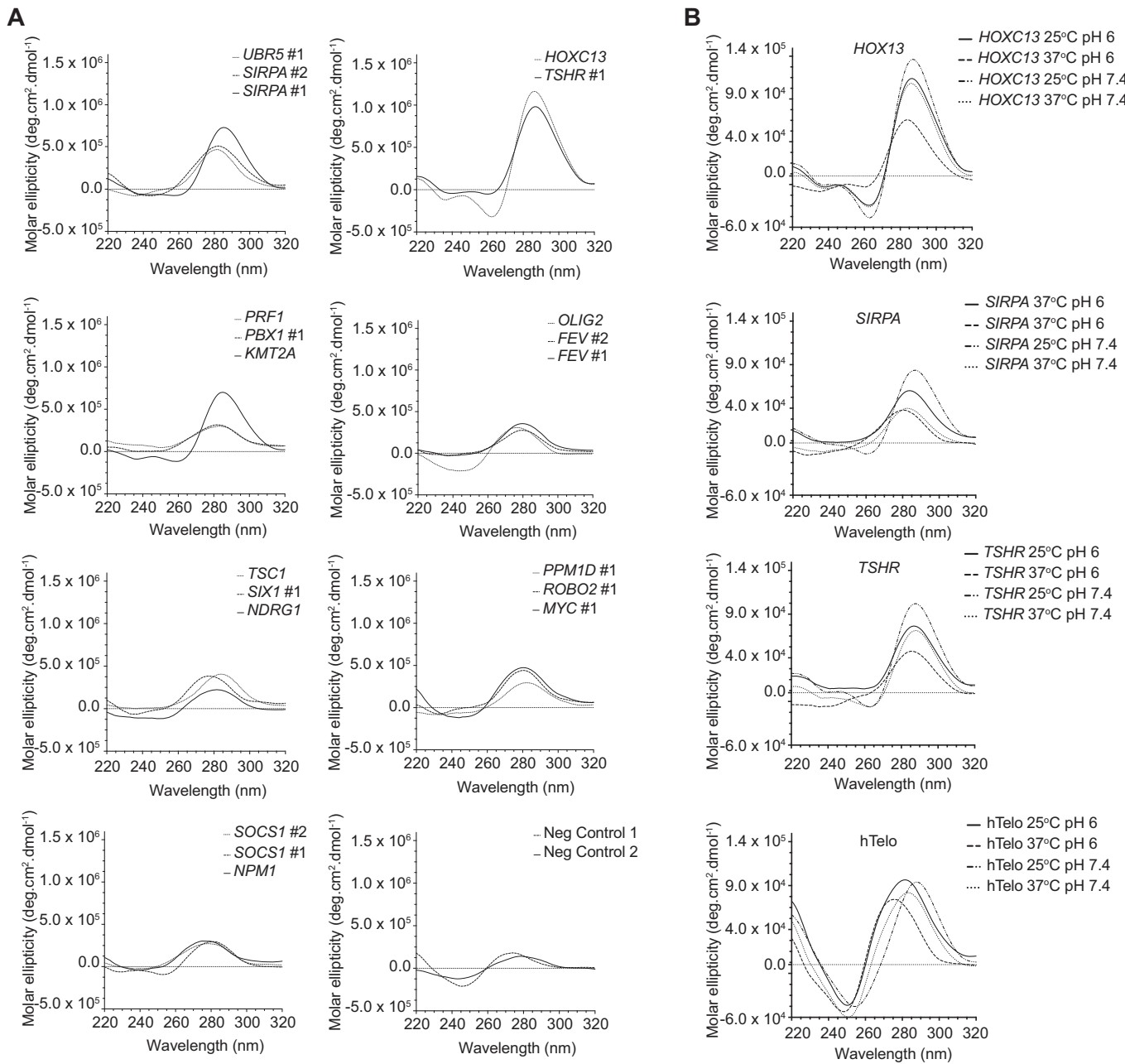

**Figure EV2. Biophysical validation of iM folding.**

(A) Validation of identified iMs by DNA synthesis and circular dichroism spectroscopy at pH 6.0 and a temperature of 25 °C of selected sequences proximal of promoter regions in known oncogenes. NC1(5'-CAGACTGTCGATGAAGCCCTG-3') and NC2 (5'-CTAGTTATTGCTCAGCGGTG-3') negative control sequences. (B) Effects of Temperature (25 °C or 37 °C) and pH (pH 6 or pH 7.4) on CD spectroscopy using identified iM regions associated to the genes *HOXC13*, *SIRPA*, *TSHR*, and the positive control hTelo sequence.

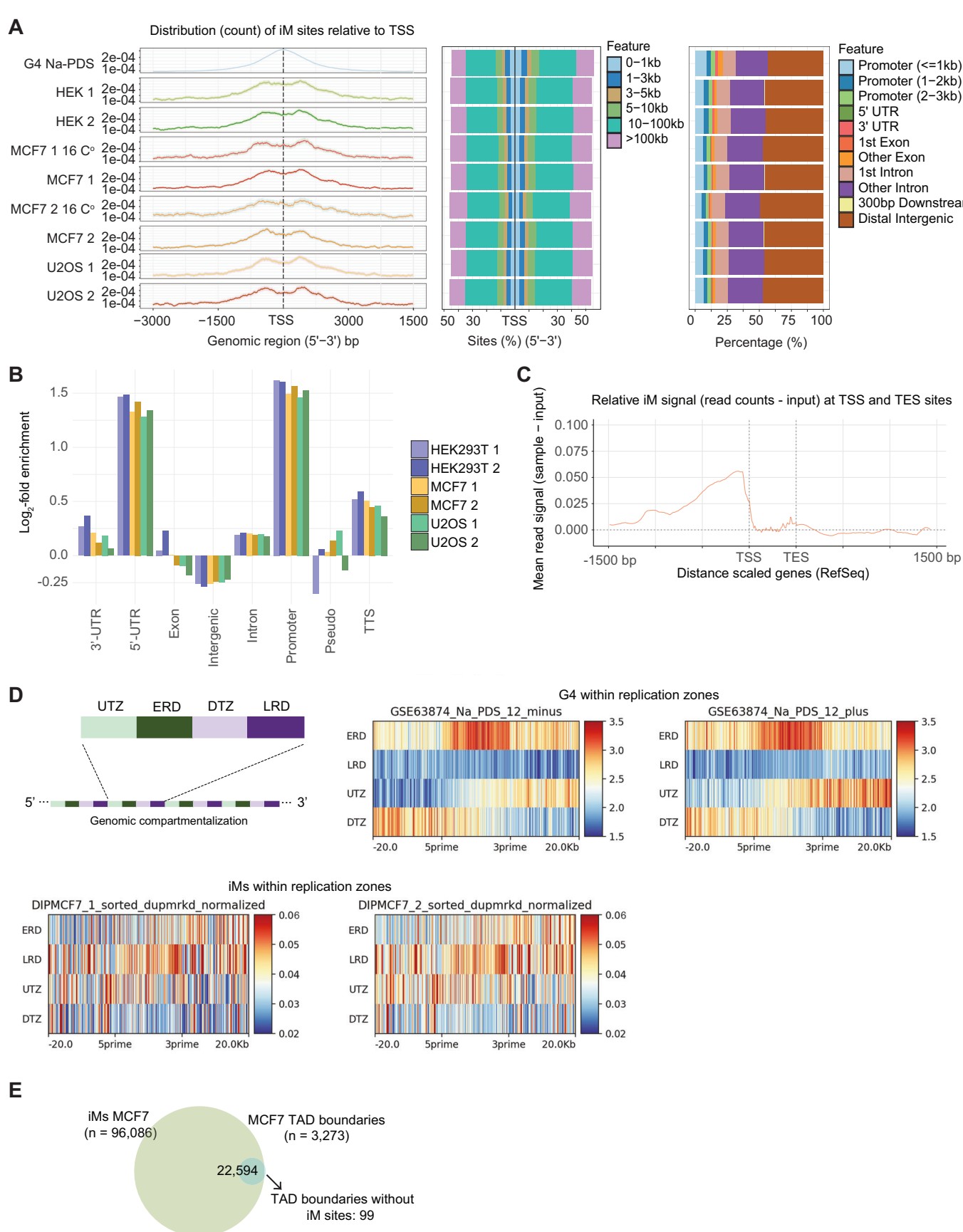

**Figure EV3. Distribution and sequences of iM regions in common genomic features, DNA replication zones and TAD boundaries.**

(A) Distribution of genomic DNA iM structures across the human genome features. Left panels show the distribution of iM pulldown relative to all TSS regions (Refseq). Centre panels show the percentage of occupancy of iM sites relative to TSS distance. Right panels represent the percentage in relationship with the common gene body features. (B) $\log_2$ fold enrichment of iM regions distributed across most common genomic annotations. (C) Relative immunoprecipitated DNA mean signal normalised (sample-input) in relationship to RefSeq gene coordinates. (D) Genomic partition of replication zones (analyses based on repetition of up transition zones (UTZ), early replication zones (ERD), down transition zones (DTZ) and late replication domains (LRD); Tag count heatmaps of G4 (Chambers et al, 2015) (upper panels) and iMs (lower panels; two biological replicates shown). (E) Overlap of iM regions found in DNA purified from MCF7 replicates and previously reported MCF7 TAD boundaries (GSE66733).

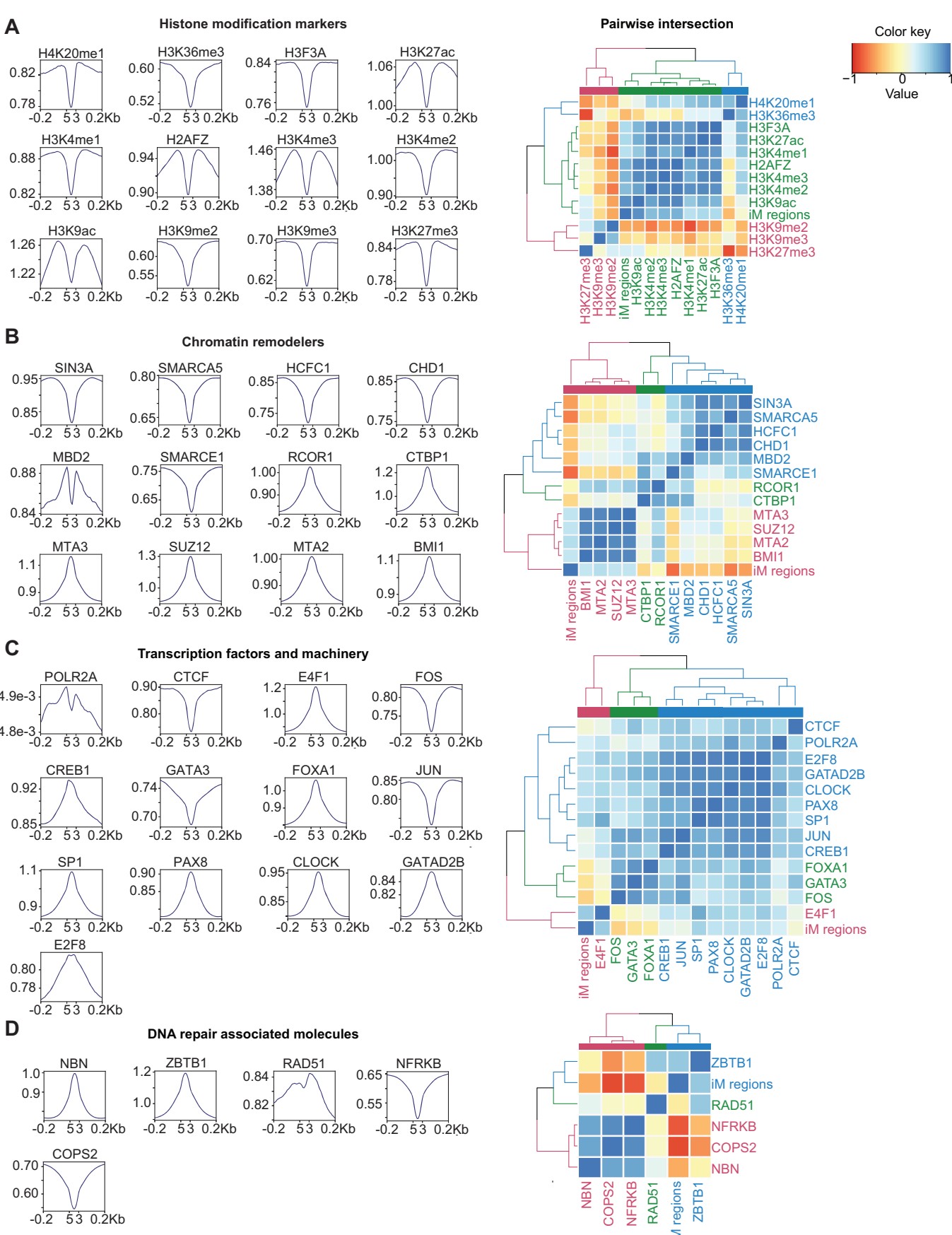

**Figure EV4. iM regions occupy transcription machinery and active transcription histone modification sites.**

Read count occupancy of iMab pulldowns relative to reported ChIP-sequencing from transcription factors and histone modifications in MCF7 cells (ENCODE) and pairwise intersection of iM regions in MCF7 cells (spearman correlation values shown). (**A**) Histone modification markers. (**B**) Chromatin remodelers. (**C**) Transcription factors and transcription-related machinery. (**D**) DNA repair-associated molecules.

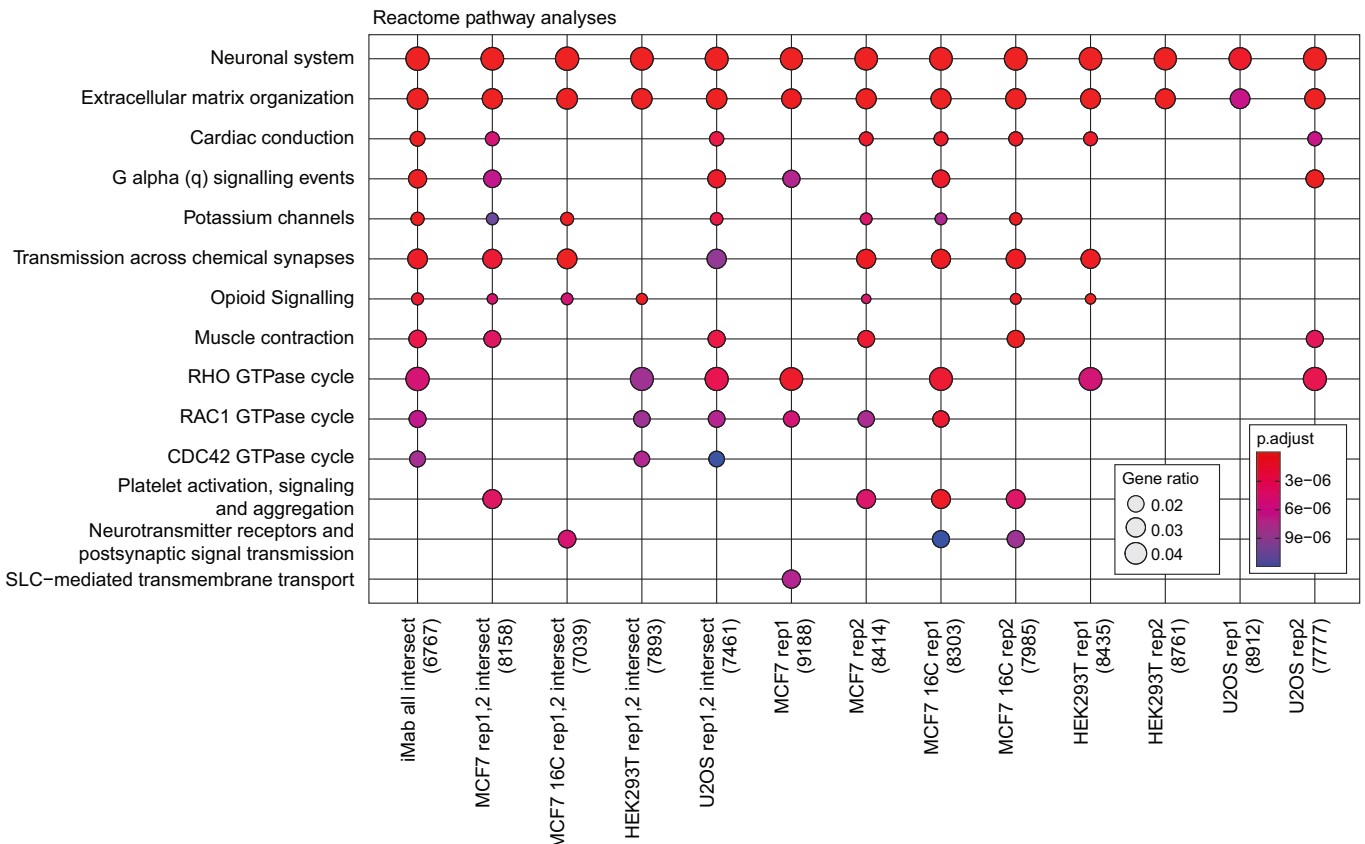

**Figure EV5.   Pathway enrichment across sample sets.**

Reactome pathway enrichment analysis across samples from the different cell-lines and conditions. Ten most significant pathways shown with *P* value adjusted and scaled to highlight ratio of genes.

