## [Peer Review File · The EMBO Journal]

Human genomic DNA is widely interspersed with i-motif structures

Cristian Peña Martínez, Mahdi Zeraati, Romain Rouet, Ohan Mazigi, Jake Henry, Brian Gloss, Jessica Kretzmann, Cameron Evans, Emanuela Ruggiero, Irene Zanin, Maja Marusic, Janez Plavec, Sara Richter, Tracy Bryan, Nicole Smith, Marcel Dinger, Sarah Kummerfeld, and Daniel Christ

Corresponding author(s): Daniel Christ (d.christ@garvan.org.au)

Review Timeline:

Submission Date:	7th Mar 24
Editorial Decision:	15th Mar 24
Revision Received:	26th Jun 24
Editorial Decision:	26th Jul 24
Revision Received:	5th Aug 24
Accepted:	9th Aug 24

Editor: Hartmut Vodermaier

Transaction Report:

This manuscript was transferred to The EMBO JOURNAL following peer review at another journal.

Dr Daniel Christ
The Garvan Institute of Medical Research
Immunology and Inflammation
384 Victoria Street
Darlinghurst Sydney, NSW 2010
Australia

26th Apr 2024

Re: EMBOJ-2024-117208-T
Human genomic DNA is widely interspersed with i-motif structures

Dear Dr Christ,

Thank you for transferring your revised manuscript to The EMBO Journal. As already communicated, we would be interested in publishing your study without further experimental revisions, as long as the remaining concerns of the original referees 1 and 3 can be satisfactorily responded to. In particular, it shall be important to clarify doubts around the iMab tool specificity that may affect the physiological significance of the work. As mentioned, we would involve a trusted arbitrator of our journal's choice to assess your final responses.

In order to allow you to update your manuscript files and to upload a detailed response letter to the last round of reviews, I am herewith sending you a formal revision invitation and a link for resubmission (below).

I am looking forward to receiving your revised version.

With kind regards,

Hartmut Vodermaier

1) Every manuscript requires a Data Availability section (even if only stating that no deposited datasets are included). Primary datasets or computer code produced in the current study have to be deposited in appropriate public repositories prior to resubmission, and reviewer access details provided in case that public access is not yet allowed. Further information:

embopress.org/page/journal/14602075/authorguide#dataavailability

9) Digital image enhancement is acceptable practice, as long as it accurately represents the original data and conforms to community standards. If a figure has been subjected to significant electronic manipulation, this must be clearly noted in the figure legend and/or the 'Materials and Methods' section. The editors reserve the right to request original versions of figures and the original images that were used to assemble the figure. Finally, we generally encourage uploading of numerical as well as gel/blot image source data; for details see: embopress.org/page/journal/14602075/authorguide#sourcedata

At EMBO Press, we ask authors to provide source data for the main manuscript figures. Our source data coordinator will contact you to discuss which figure panels we would need source data for and will also provide you with helpful tips on how to upload and organize the files.

Further information is available in our Guide For Authors:

In the interest of ensuring the conceptual advance provided by the work, we recommend submitting a revision within 3 months (25th Jul 2024). Please discuss the revision progress ahead of this time with the editor if you require more time to complete the revisions. Use the link below to submit your revision:

Link Not Available

Response to Reviewers' Comments (2nd revision):

We thank the reviewers for their extensive comments. Please find our detailed point-by-point response to their comments, as well as a revised manuscript as attached. We have now significantly expanded the discussion and extensively discuss recent publications, as well as the pre-print by Boissieras *et al.* We have also extensively re-written the conclusion to further address potential and limitations of the approach and to discuss recent publications.

Major comment 1: Response to the pre-print of Boissieras *et al.*

Reviewer 3: Finally, a new preprint by Boissieras et al. (<https://doi.org/10.1101/2023.11.21.568054>) appeared providing evidence that the iMab antibody, used by the authors here to infer iM formation genome-wide by iM-DIP-seq, binds single-stranded C-rich sequences and actively unfolds iM structures to C-rich ssDNA. The authors should consider these results and comment whether their iM-DIP-seq approach observed genomic locations that could form iM structures in vivo, or whether it is unclear whether they mapped genomic regions forming C-rich ssDNA or iM structures.

Our response:

We considered it likely that the pre-print by Boissieras et al. (despite being non-peer reviewed) had negatively influenced the reviewer base and would require an experimental response. As the iMab antibody is widely used in the field we were contacted by several colleagues (Plavec, Richter), and have jointly and independently repeated the experiments described by Boissieras et al.. This analysis revealed that several of the oligonucleotides used by Boissieras *et al.* form *intermolecular* iM structures at higher DNA concentrations, which was further confirmed by NMR studies, providing a direct explanation for the observed discrepancies reported by the authors. In contrast, highly iM-specific pulldowns were observed at lower DNA concentrations in the physiological range, highlighting the specificity of the iMab antibody, which recognises both intra- and intermolecular iMs.

Reference: Ruggiero, E.M., M; Zanin, I; Peña Martinez, C.D.; Plavec, J; Christ, D; Richter, S.N. , *The iMab antibody selectively binds to intramolecular and intermolecular i-motif structures*. BIORXIV/2024/600195, 2024.

We now extensively discuss this topic in the manuscript:

In contrast to the published studies as above, a recent pre-print by Boissieras *et al.* using *in vitro* measurements reported that the iMab antibody binds to C-rich sequences independently of iM formation [51]. This contrasts with the study by Zanin *et al.* (as well as the data outlined here), which demonstrate by CD that the identified sequences predominately fold into iM structures [16]. Similar observations were also made by Ma *et al.* who reported that all the DNA sequences that were randomly selected from their set of 25,306 hits, and characterized by CD, formed iM structures [49] (the study was not referenced by Boissieras *et al.*).

To further investigate the reported discrepancies, Ruggiero *et al.* have recently performed pulldown experiments using iMab and the set of synthetic oligonucleotides used by Boissieras *et al.* [52]. This analysis revealed that several of the oligonucleotides used by Boissieras *et al.* form *intermolecular* iM structures at higher DNA concentrations, which was further confirmed by NMR studies, providing a direct explanation for the observed discrepancies reported by the authors [52]. In contrast, highly iM-specific pulldowns were observed at lower DNA concentrations in the physiological range, highlighting the specificity of the iMab antibody, which recognises both intra- and intermolecular iMs [52].

C: Pull-down/WB performed with low/high DNA concentrations

Point-by-point response Figure 1 (with S. N. Richter (Italy) and J. Plavec (Slovenia): Oligonucleotides used by **Boissieras et al.** form *intermolecular* iM structures which are specifically recognized by the iMab antibody providing a direct molecular explanation for the reported discrepancies; pulldowns at low DNA concentrations, under which *intermolecular* iM do not form, overcome these artefacts. A) Imino region of 1D ^1H NMR spectra recorded at 5 °C, at different pH and at oligonucleotide concentrations. Regions characteristic for signals of protons included in non-canonical C-C⁺ and T-T base-pairs are highlighted by the yellow and grey areas, respectively. B) $T_{1/2}$ at two different oligonucleotide concentrations obtained from NMR melting experiments based on the intensity of the signals in the imino region. Data points in the melting profiles displayed in grey represent data at 0.1 mM oligonucleotide concentration, while data points in shades of blue represent data at 1.0 mM oligonucleotide concentration. C) Pull-down/WB performed with low (300 ng) and high (1500 ng) DNA amounts, with 10 ng iMab per sample. The iMab antibody specifically recognizes *intramolecular* iM structures at low concentrations, while at (unphysiologically) high DNA concentration *intermolecular* iM formation is also observed, which are also specifically recognized by iMab.

Major comment 2: Response to condition used for immunoprecipitation and biophysical validation

Reviewer 1: I wonder whether the Authors would recapitulate the iM genome-wide profiles obtained by performing the immunoprecipitation step on naked DNA if they used the same cell fixation conditions as in Zeerarati et al but then performed iM immunoprecipitation and sequencing instead of immunofluorescence.... I wonder if heat shock would destabilize some of the iM structures. It would be great if the Authors could identify some in vivo perturbations able to reshape the genomic iM landscape in a reversible manner. Reviewer 3: 37C is the physiologically relevant temperature of human cells and that iM folding is dependent on temperature and pH, requires an adjusted iM-DIP seq protocol that contains either 37C at the immunoprecipitation step or 37C washing steps.

Our response:

The conditions and temperatures used here are standard immunoprecipitation conditions that are also used by others in the field, including the two seminal studies by Ma et. al and Lam et al. Using immunofluorescence or heat shock conditions (as suggested by Reviewer 1) or 37°C (as suggested by Reviewer 2) is not common. We on purpose utilized standard IP conditions to allow direct comparisons to with the two prior seminal studies by Ma and Lam. I am also not convinced that Reviewer 1 is aware that the Zeraati et al study was published by my group and that we are therefore well-aware of the conditions used. However, we do agree with the reviewers that the use of protein-depleted DNA and 4°C are *general* limitations of IP approaches and now specifically discuss these limitations in the paper:

It is important to note that the observed differences are likely to partially reflect different experimental procedures, conditions and reagents. Thus, immunoprecipitation studies including the studies by Ma et al. [49] and the study described here are carried out at 4°C, while live cell studies are generally carried out 37°C [14, 16, 17]. Moreover, different conditions were used: in particular, strong blocking conditions (milk powder, superbloc, spermidine, high ionic strength, salmon sperm DNA) were used here and by the above studies but not by Boissieras et al [51], which might well have contributed to the non-specificity reported by the authors [52]. Specific antibody formats used in the different studies may have also contributed to the observed differences, with some of the studies (including Ma et al.) [49] utilizing an IgG format, while other studies predominately used iMab in a single chain Fv (scFv) antibody format.

While differences in experimental approaches (including the use of rice DNA and acidic conditions (pH 5.5)) exclude direct comparisons, the studies by Ma et al and Zanin et al highlight the potential of iMab, developed in our laboratory, for immunoprecipitation and tagmentation approaches across a wide range of organisms. In contrast to the tagmentation studies above [16], results outlined here display a high level of consistency of iM regions among three different human cells lines (Fig 1b), indicating experimental robustness and a high degree of iM convergence between cells of different morphological origins, including

cancer cell lines. On the other hand, the iM mapping performed in the chromatin context, revealed different iM distribution among cell lines [16], in line with the observation that the chromatin dynamic architecture influences cell identity [53]. Although the study outlined here utilized protein-depleted human genomic DNA, we observed overall similar results with the study of Zanin et al. [16]: in both cases, iM structures were observed located in close proximity to G4 forming regions and in genes with high transcription rates, indicating that iM and G4 forming sequences in the human genome are mostly observed within regions that regulate gene expression.

Ma et al:

DNA was diluted with iM-IP incubation buffer (50 mM Tris-AcOH, 1 mM MgCl₂, 130 nM CaCl₂, 1% BSA and Complete mini, pH 5.5), then incubated with 3 µg iMab antibody (Ab01462-23.0, Kappa) for 4 h at 4°C. The antibody incubation reaction was incubated with 30 µl of washed protein G Dynalbeads (10004D, Invitrogen) for another 4 h at 4°C

Ma, X., et al., *Genome-wide characterization of i-motifs and their potential roles in the stability and evolution of transposable elements in rice*. Nucleic Acids Res, 2022.

Lam et al.

Beads were washed three times with 0.5% BSA then incubated with 400 µl of 300 ng µl⁻¹sonicated genomic DNA. Following overnight incubation rotating at 4 °C, beads were washed six times with 10 mM Tris pH 7.4, 100 mM KCl, 0.1% tween then once with 10 mM Tris pH 7.4, 100 mM KCl. Bound DNA was eluted in 50 µl of 1% SDS, 0.1 M NaHCO₃ at 30 °C for 1 h then purified with Roche PCR purification columns

Lam, E.Y., et al., *G-quadruplex structures are stable and detectable in human genomic DNA*. Nat Commun, 2013. 4: p. 1796.

Other comments:

Reviewer 1: The Authors show an enrichment of iM structures around the TSS but it is unclear whether this is just the result of these regions being C-rich or whether the formation of iMs is related to the transcriptional activity of the genes. The Authors should examine the enrichment of iMs around TSS for different groups of genes based on their expression levels (e.g., for each expression quantile).

Our response:

This is a valid question. However, we would consider this outside the scope of this experimental study, and more suited for future computational analyses. Indeed, the provision of a foundational experimental resource to the community for this type of future analysis is a major aim of this manuscript.

Reviewer 3: CD experiments at 37C pH 7.4 have not addressed whether endogenous genomic DNA can form iM structures at 37C pH 7.4

Our response:

We re-iterate that it is essential to validate the formation of iM structure in the absence of antibody (as we have carried out by CD at both 25°C and 37°C— see also the recent studies Zanin et al and Ma. et al – now extensively discussed in the manuscript - both use our approach and confirm our findings). As we had pointed out in our initial response, reviewer 3 does not comprehend that the binding energy of the iMab antibody (nanomolar affinity) is in the range of the folding energy of the iM structure (at 10-15 kcal/mol). This has important implications, as the overall iM folding energy ($\Delta G^\circ \text{ total} = \Delta G^\circ \text{ folding} + \Delta G^\circ \text{ binding}$) is dominated by the affinity component not the temperature component of the equation. This also applies to other high affinity antibodies such as BG4. It is therefore essential to validate iM folding in the *absence* of antibody. Washing the antibody/DNA complex at 37°C is not a suitable proxy, as this reviewer seems to assume, as a high affinity antibody will stabilize the complex under such conditions. Please contact an expert in biophysics and antibody-binding in case there are outstanding questions. Recommended: Alan Fersht arf25@cam.ac.uk or David Baker dabaker@uw.edu.

We now specifically mention in the manuscript, and have included a new Extended data figure 1f (which previously had only been included in the point-by-point response to reviewers):

Importantly, folding was observed at both 25°C and at 37°C, indicating the potential of the identified sequences to fold at physiological temperatures (Extended data Fig. 1f).

Prof. Daniel Christ
The Garvan Institute of Medical Research
Immunology and Inflammation
384 Victoria Street
Darlinghurst Sydney, NSW 2010
Australia

26th Jul 2024

Re: EMBOJ-2024-117208R1
Human genomic DNA is widely interspersed with i-motif structures

Dear Dr. Christ,

Thank you again for transferring your revised manuscript, together with referee reports from a previous journal, to The EMBO Journal. I apologize for the delay in getting back to with a response, as I had been away from the office for an extended period. I have now had the chance to carefully assess your responses to the final round of referee comments, and found your arguments as well as the supporting data included in here and in your new preprint well-taken and convincing. We therefore decided that we would be happy to accept the study without additional (arbitrating) referee input at this point.

Before we can proceed with formal acceptance, I would now still need to ask you to reformat the manuscript according to EMBO Journal guidelines (see <https://www.embopress.org/page/journal/14602075/authorguide>), and to diligently address the following editorial issues:

- Please enter every author on the manuscript into our submission system, specifying their particular contributions formally based on Contributor Role Taxonomy (CRediT) terms directly in the Author Information page of our submission system (this replaces informal Author Contribution statements in the text - see <https://casrai.org/credit/> for more information).
- Please reorganize the figures in order to better capitalize on The EMBO Journal's full article format. We usually include more than three main figures, so you may include some of your supplementary data in main figures. We also allow up to five Expanded View figures (naming/citation: 'Figure EV1-5') that will also be typeset and directly accessible in the HTML version of the published article; their legends should therefore also be part of the main text. Finally, the table containing BLI/CD oligonucleotides should be renamed and referenced as 'Table EV1', and its title & legend moved from the main text into a separate 'Legend' tab of the the XLSX spreadsheet.
- Please upload all main Figures and all Expanded View figures as individual files with sufficient resolution/quality for production.
- Please adjust the order of the manuscript sections: Title page with complete author information, Abstract, Keywords, Introduction, Results, Discussion, Materials & Methods, Data Availability Section, Acknowledgements, Disclosure and Competing Interests Statement, References, Main figure legends, Tables, Expanded Figure Legends.
- On the abstract page of the manuscript, please include 4-5 general keyword terms to enhance searchability.
- Please use the header 'Introduction' (rather than 'Main') for the section between abstract and results, and make sure to expand it to better introduce the background of this research to a broad readership - there are no length restrictions in EMBO Journal articles.
- Please adjust the format of the reference list and of the in-text citations according to EMBO Journal format (alphabetical order, author name et al + year...). Please also note the specific format for citation of preprints as specified in our author guidelines: The citation in the text should be: "(preprint: NAME1 et al, YEAR)"
The citation in the reference list: "Author NAME1, Author NAME2, ... (YEAR) article title. bioRxiv/ResearchSquare doi: XXX"
- All Materials and Methods need to be described in the main text using our 'Structured Methods' format. The Methods section should include a Reagents and Tools Table (downloadable at <https://www.embopress.org/page/journal/14693178/authorguide#structuredmethods>) listing key reagents, experimental models, software and relevant equipment, and including their sources and relevant identifiers; followed by a Methods and Protocols section describing the methods (ideally using a step-by-step protocol format to facilitate adoption of the methodologies across labs)
- Please add an Acknowledgement section (if appropriate) listing funding information, and make sure to enter the same funding information also in our submission system.

- Please include a Disclosure and competing interests statement (next to the Acknowledgment section) - for details, see <https://www.embopress.org/competing-interests>

- In the Data Availability section, please include direct URLs for the databases (only GEO here?) in which the NEWLY GENERATED data have been deposited. Previously generated datasets used in the present study should instead be referenced in the appropriate section of the Methods. Please note that we encourage the use of formal 'data citation'/data references' (see <https://www.embopress.org/page/journal/14602075/authorguide#referencesformat> for explanation); but you may also only list accession codes and the respective primary publication (esp. for the case of all the ENCODE project entries that you listed, this may be the easiest solution).

- In the Author Checklist, please enter the full corresponding author name, as well as the correct journal name.

- Finally, please provide suggestions for a short 'blurb' text prefacing and summing up the study in two sentences (max. 250 characters), followed by 3-5 one-sentence 'bullet points' with brief factual statements of key results of the paper; they will form the basis of an editor-written 'Synopsis' accompanying the online version of the article. Please also upload a synopsis image, which can be used as a "visual title" for the synopsis section of your paper. The image should be in PNG or JPG format, and please make sure that it remains in the modest dimensions of (exactly) 550 pixels wide and 300-600 pixels high.

I am therefore inviting you to a final round of formal revision, solely to allow you to make these modifications and upload all revised files. Once we will have received them, we should hopefully be ready to swiftly proceed with publication of the study!

With best regards,

Hartmut

*** PLEASE NOTE: All revised manuscripts are subject to initial checks for completeness and adherence to our formatting guidelines. Revisions may be returned to the authors and delayed in their editorial re-evaluation if they fail to comply to the following requirements (see also our Guide to Authors for further information):

1) Every manuscript requires a Data Availability section (even if only stating that no deposited datasets are included). Primary datasets or computer code produced in the current study have to be deposited in appropriate public repositories prior to resubmission, and reviewer access details provided in case that public access is not yet allowed. Further information: [embopress.org/page/journal/14602075/authorguide#dataavailability](https://www.embopress.org/page/journal/14602075/authorguide#dataavailability)

6) Please complete our Author Checklist, and make sure that information entered into the checklist is also reflected in the manuscript; the checklist will be available to readers as part of the Review Process File. A download link is found at the top of our Guide to Authors: [embopress.org/page/journal/14602075/authorguide](https://www.embopress.org/page/journal/14602075/authorguide)

7) All authors listed as (co-)corresponding need to deposit, in their respective author profiles in our submission system, a unique

ORCID identifier linked to their name. Please see our Guide to Authors for detailed instructions.

9) To facilitate reproducibility and cross-laboratory adoption of methodologies, please structure the Materials & Methods section as outlined in our guide to authors, including a completed Reagents and Tools Table that can be downloaded from our author guidelines as well (<https://www.embopress.org/page/journal/14602075/authorguide#structuredmethods>).

10) Digital image enhancement is acceptable practice, as long as it accurately represents the original data and conforms to community standards. If a figure has been subjected to significant electronic manipulation, this must be clearly noted in the figure legend and/or the 'Materials and Methods' section. The editors reserve the right to request original versions of figures and the original images that were used to assemble the figure. Finally, we generally encourage uploading of numerical as well as gel/blot image source data; for details see: embopress.org/page/journal/14602075/authorguide#sourcedata

At EMBO Press, we ask authors to provide source data for the main manuscript figures. Our source data coordinator will contact you to discuss which figure panels we would need source data for and will also provide you with helpful tips on how to upload and organize the files.

In the interest of ensuring the conceptual advance provided by the work, we recommend submitting a revision within 3 months (24th Oct 2024). Please discuss the revision progress ahead of this time with the editor if you require more time to complete the revisions. Use the link below to submit your revision:

Link Not Available

Prof. Daniel Christ
Garvan Institute of Medical Research
Immunology and Inflammation
384 Victoria Street
Darlinghurst Sydney, New South Wales 2010
Australia

9th Aug 2024

Re: EMBOJ-2024-117208R2
Human genomic DNA is widely interspersed with i-motif structures

Dear Prof. Christ,

Thank you for submitting your final revised manuscript for our consideration. I am pleased to inform you that we have now accepted it for publication in The EMBO Journal.

Yours sincerely,

Hartmut Vodermaier
